# GL-NeRF: Gauss-Laguerre Quadrature Enables Training-Free NeRF Acceleration

**Silong Yong**    **Yaqi Xie**    **Simon Stepputtis**    **Katia Sycara**
Carnegie Mellon University
{silongy, yaqix, sstepput, sycara}@andrew.cmu.edu

## Abstract

Volume rendering in neural radiance fields is inherently time-consuming due to the large number of MLP calls on the points sampled per ray. Previous works would address this issue by introducing new neural networks or data structures. In this work, we propose GL-NeRF, a new perspective of computing volume rendering with the Gauss-Laguerre quadrature. GL-NeRF significantly reduces the number of MLP calls needed for volume rendering, introducing no additional data structures or neural networks. The simple formulation makes adopting GL-NeRF in any NeRF model possible. In the paper, we first justify the use of the Gauss-Laguerre quadrature and then demonstrate this plug-and-play attribute by implementing it in two different NeRF models. We show that with a minimal drop in performance, GL-NeRF can significantly reduce the number of MLP calls, showing the potential to speed up any NeRF model. Code can be found in project page https://silongyong.github.io/GL-NeRF_project_page/.

## 1   Introduction

Neural Radiance Fields (NeRFs) [27] have shown promising results for synthesizing images from novel views. Plenty of works extend NeRF towards different aspects applicable in the real world (see related works for details). The core component for NeRF's success is volume rendering, which requires approximating an integral by densely sampling points along the ray and evaluating volume density and radiance using neural networks for them. In practice, a dense set of points is evaluated by expensive operations like neural network inferences for a single pixel, which could be redundant. Works have been done to reduce the time needed for rendering images, aiming at providing NeRF with a real-time rendering ability [9, 44, 24, 29, 8]. Despite the promising results shown by these works, they propose different approaches for achieving real-time rendering by introducing new networks, new data structures, *etc*. Therefore, each individual work requires training from scratch with a specific optimization goal. In this work, we propose a novel lightweight method that could be implemented in any existing NeRF-based models that require volume rendering without further training. In contrast to existing works, our approach introduces no additional representation or neural network and is training-free. We make minimal modifications to the computation of the volume rendering integral, making it rely on much fewer samples.

Our approach arises from revisiting the volume rendering integral, the key discovery is that with a simple change of variable, we can turn the integral into a pure exponentially weighted integral of color. This specific form has a Gauss quadrature (*i.e.* the Gauss-Laguerre quadrature) which best approximates it mathematically. Naturally, we propose to use the Gauss-Laguerre quadrature to directly compute the volume rendering integral, which we call GL-NeRF (Gauss Laguerre-NeRF), leading to much lower computational cost for approximating the integral and therefore lower time and memory usage. Computing the points needed for the integral requires a dense evaluation of per-point density. However, the efficiency for this step can be improved using modern techniques

38th Conference on Neural Information Processing Systems (NeurIPS 2024).

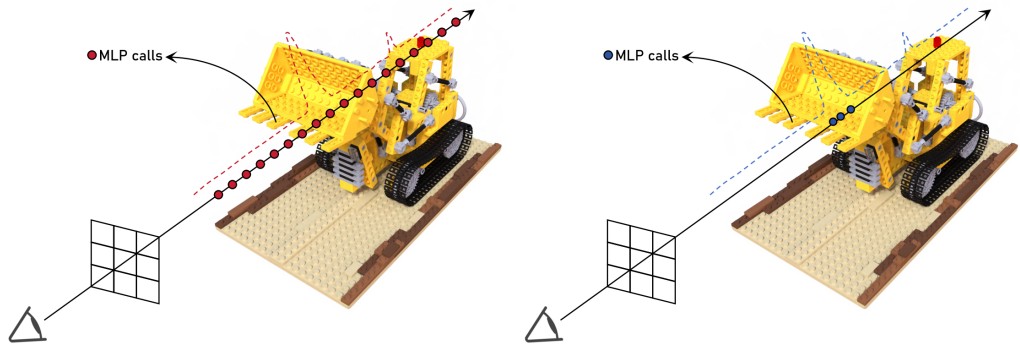

MLP calls ←

NeRF uniform sample rendering

MLP calls ←

Gauss-Laguerre Quadrature rendering (ours)

Figure 1: GL-NeRF method overview. The vanilla volume rendering in NeRF requires uniform sampling in space. This leads to a huge number of computationally heavy MLP calls since we have to assign each point a color value. Our approach, GL-NeRF, significantly reduces the number of points needed for volume rendering and selects points in the most informative area.

like factorized tensors [6]. Benefiting from the guarantee of the highest precision Gauss quadrature provides, only a very small number of fixed points could provide comparable results to the heavy and redundant strategy NeRF adopts, leading to free speedup.

To verify the use of the Gauss-Laguerre quadrature, we conduct an empirical study on the landscape of color function. We also analyze the relationship between our approach and other techniques that aim to reduce the sample points for NeRF. We demonstrate the plug-and-play property of our method by directly incorporating it into vanilla NeRF and TensoRF models that are already trained on NeRF-Synthetic and LLFF datasets. Furthermore, we showcase the drop in time and memory usage as a direct outcome of reducing the computational cost.

GL-NeRF provides a different perspective for computing volume rendering and has the potential to be a direct plug-in for existing NeRF-based products. Specifically, our contributions are three-fold. We propose GL-NeRF, a brand new perspective for computing volume rendering with the Gauss-Laguerre quadrature with no additional component introduced. We analyze the validity of using the Gauss-Laguerre quadrature for volume rendering integral and the relationship between our approach and existing sample-efficient NeRFs. We demonstrate that GL-NeRF could be incorporated into any NeRF model without further training. To the best of our knowledge, GL-NeRF is the first method that could be used without training in any NeRF models thanks to the simple formulation. We showcase that GL-NeRF reduces computational cost, time and memory usage while keeping the rendering quality.

## 2 Related work

**Volume rendering.** Volume rendering has been widely used in computer graphics and vision applications [25, 43, 7]. It maps a 3D scene onto 2D images by a weighted integral over the color of the points along the corresponding rays with a function of opacity (volume density) as weight. In practice, the integral is approximated using a finite sum over sampled points along the ray as derived in [25]. Implicit scene models like NeRF [27], Plenoxels [8] and 3D gaussians [18] and most of their follow-up all adopt this technique as the render pipeline. Since randomly sampling in space for approximating the integral may bring unnecessary information (*i.e.* sampling in empty space) that may cost extra computation, plenty of works aim to address that by introducing different techniques for better approximation of the component needed for volume rendering integral (*i.e.* volume density, radiance) [40, 44, 29, 23, 21, 2, 36]. PL-NeRF [40] proposes to use piecewise linear function for approximating the volume density throughout the space, leading to fewer points needed for the "fine" stage sampling proposed by [27]. AutoInt and DIVeR [23, 44] introduce a neural network for approximating the integral of volume density instead of using Monte-Carlo sampling. DONeRF [29] reduces the sampled point needed for computing the integral by introducing a depth oracle neural network that predicts the surface position of the underlying scene and samples the points near the

**Algorithm 1** Gauss Laguerre Quadrature for Volume Rendering

---

**Input:** ray direction $d$, ray origin $o$, step size $\Delta t$, *sample number* M, *Gauss-Laguerre quadrature weight look-up table* $L_w$, *point look-up table* $L_p$

1: $t_{\min}, t_{\max}$ = RayIntersectBoundingBox($d, o$)
2: **if** $t_{\min} > t_{\max}$ **then return** `bg_color`
3: Initialize $t = t_{min}$, transmittance $T = 1.0$, already sampled point number $n = 0$
4: **while** $t < t_{\max}$ **do**
5:      **if** $(n == m)$ **then** $break$
6:      `pos` $= o + t * d$
7:      $\sigma$ = GetVolumeDensity(`pos`)
8:      $x = -log(T)$
9:      $x_{\text{next}} = x + \Delta t * \sigma$
10:      **if** $x < L_p[n]$ and $x_{\text{next}} \geq L_p[n]$ **then**
11:          $t_{\text{Laguerre}} = (L_p[n] - x)/(x_{\text{next}} - x)$
12:          $\text{pos}_{\text{sample}} = o + (t + t_{\text{Laguerre}} * \Delta t - \Delta t) * d$
13:          `ray_color`$+ = L_w[n] *$ GetColor($\text{pos}_{\text{sample}}$)
14:          $n = n + 1$
15:      $t = t + \Delta t$
16:      $T = T * exp(-\sigma * \Delta t)$
17: `bg_weight` $= sum(L_w[n :])$
18: `ray_color` $=$ `ray_color` $+$ `bg_weight` $*$ `bg_color`
19: **return** `ray_color`

---

surface, which contributes the most to the visual effect in the images. MCNeRF [13] proposes to use Monte-Carlo rendering and denoising to do sample efficient rendering, but it still introduces a denoiser network that requires per-scene training. Different from these previous works, Our work proposes to use the Gauss-Laguerre quadrature to directly improve the precision of the volume rendering integral itself, introduces no additional neural networks or data structures and remains in the simplest version, leading to its adaptability into any existing work that relies on volume rendering integral.

**NeRFs.** Neural Radiance Fields (NeRFs) have proved to be a powerful tool for novel view synthesis [27]. It uses a coordinate-based multi-layer perceptron (MLP) to represent the scene and render high-fidelity images from different views. The render is done by pixel-wise volumetric rendering [25] with density and color evaluated using the MLP on hundreds of sampled points along the ray. For modeling high-frequency information in the scene, NeRF uses positional encoding to map the input coordinates onto high-frequency bands. The success of NeRF has triggered an explosive emergence of follow-up works. There are plenty of works focusing on improving or extending the ability of NeRF towards different aspects. Aliasing along xy coordinates has been tackled [3], unbounded scenes [4, 47, 39, 34], dynamic scenes [32, 22, 30] and scenes with semantic information [41, 37, 49, 19] have been well explored and demonstrated the potential of implicit scene representation with NeRF. Nonetheless, NeRF requires plenty of time for training and rendering, blocking its way of being used for real-time rendering. The bottleneck of the computation time is the MLP used. There are two main branches of work for extending NeRF towards real-time rendering. The first branch introduces different data structure [46, 9, 14, 33, 8, 6] for scene representation. Another branch, in which our method falls, improves the sample efficiency of the model [20, 29, 31, 38] to accelerate NeRF rendering process. While previous works draw their intuition from the underlying physics perspective and thus need different formulations of the sampling strategy and different neural network architecture for predicting the surface position of the underlying scenes, we propose our method based on a mathematical observation while maintaining the overall pipeline. Benefiting from this, our work could be seamlessly incorporated into any existing NeRF-related works without further training. On the other hand, despite being derived from the mathematical perspective, our method still intuitively satisfies the underlying physical constraints.

## 3 Preliminaries

### 3.1 NeRF and volume rendering

NeRF [27] is a powerful implicit 3D scene model for novel view synthesis. At the core of its rendering ability is volume rendering. NeRF uses coordinate-based MLP to encode the scene, assigning volume density (opacity) and radiance (color) to spatial points. When used for synthesizing new views, it casts a ray $\boldsymbol{r}(t) = \boldsymbol{o} + t\boldsymbol{d}$ through the pixel to be rendered, sample points along the ray and compute volume density and radiance for these points. These values are then aggregated together using Eq.1 to give the color of the pixel.

$$\hat{\boldsymbol{C}}(\boldsymbol{r}) = \sum_{i=1}^{N} w_i \boldsymbol{c}(\boldsymbol{r}(t_i)), \tag{1}$$

where

$$w_i = T_i(1 - exp(-\sigma(\boldsymbol{r}(t_i))\delta_i)), \tag{2}$$

$$T_i = exp(-\sum_{j=0}^{i-1} \sigma(\boldsymbol{r}(t_j))\delta_j), \tag{3}$$

$t_i$ represents the sampled position along the ray and $\delta_i = t_{i+1} - t_i$ is the distance between two nearby sampled points. NeRF uses an MLP to represent volume density $\sigma$ and color $\boldsymbol{c}$. The loss function for training NeRF is simply the square error between rendered pixel colors and the corresponding pixel colors over batch of rays $\mathbf{R}$. Variants of NeRF like TensoRF[6] use different representations for volume density and color, but the process of volume rendering remains the same.

$$\mathbf{L} = \sum_{\boldsymbol{r} \in \mathbf{R}} \|\hat{\boldsymbol{C}}(\boldsymbol{r}) - \boldsymbol{C}(\boldsymbol{r})\|_2^2 \tag{4}$$

### 3.2 Gauss quadrature

An $n$-point Gauss quadrature [10] is a method for numerical integration that guarantees to yield exact results for integral of polynomials of degree $2n - 1$ or less, which is the highest possible precision for approximating an integral by quadrature. Intuitively, consider approximating an integral using quadrature as in Eq. 5

$$\int_{-1}^{1} p(x)dx = \sum_{i=1}^{n} w_i p(x_i), \tag{5}$$

where $p(x)$ is a polynomial of degree $2n - 1$, $w(x)$ is a weight function and $I$ is the interval for computing the integral. We first give the definition of orthogonality of two polynomials $p_m(x)$ and $p_n(x)$

$$\int_{-1}^{1} p_m(x)p_n(x)dx = 0, \tag{6}$$

where $p_m(x)$ is of degree $m$, $p_n(x)$ is of degree $n$ and $m \neq n$. we can use long division for $p(x)$ to obtain

$$p(x) = q(x)L_n(x) + r(x), \tag{7}$$

where $L_n(x)$ is a polynomial of degree $n$ that is orthogonal to any polynomials that have degree less than $n$ (*i.e.* $n$ degree Legendre polynomial), $q(x)$ and $r(x)$ are both polynomials with degree less than $n$. Then

$$\int_{-1}^{1} p(x)dx = \int_{-1}^{1} q(x)L_n(x)dx + \int_{-1}^{1} r(x)dx. \tag{8}$$

Since $L_n(x)$ is orthogonal to any polynomials with degree less than $n$, the first term on the right hand side of Eq. 8 should equal to $0$. Since it doesn't contribute to the computation of the integral, we may also neglect it when computing the quadrature. Therefore, we should choose $x_i$ that satisfies $L_n(x_i) = 0$ [16]. With this intuition bearing in mind, carefully choosing the weights $w_i$ for computing the quadrature would help us precisely calculate Eq. 5 because we have $n$ points to compute the second term on the right hand side of Eq. 8, which is an integral of a polynomial of degree less than $n$.

In general, given a function $f(x)$, Gauss quadrature computes its integral on $[-1, 1]$ using

$$\int_{-1}^{1} f(x)dx \approx \sum_{i=1}^{n} w_i f(x_i),$$

(9)

where $x_i, i = 1, 2, \ldots, n$ corresponds to a root of the orthogonal polynomials on $[-1, 1]$. This quadrature is called Gauss-Legendre quadrature since the orthogonal polynomials on $[-1, 1]$ with a weight function $g(x) = 1$ are Legendre polynomials. An $n$-th degree Legendre polynomial takes the form [35, 15, 17]

$$P_n(x) = \frac{1}{2^n n!} \frac{d^n}{dx^n} (x^2 - 1)^n,$$

(10)

and $w_i$ is computed using Eq. 11 as shown in [1].

$$w_i = \frac{2}{(1 - x_i^2)[P_n'(x_i)]^2}.$$

(11)

**Gauss-Laguerre quadrature** is an extension of Gauss quadrature for approximating integrals following the form of

$$\int_0^{\infty} e^{-x} f(x)dx \approx \sum_{i=1}^{n} w_i f(x_i).$$

(12)

In this case, the weight function is $g(x) = e^{-x}$, the integral interval is $[0, \infty)$. $x_i$ corresponds to the root of Laguerre polynomials

$$L_n(x) = \frac{1}{n!} (\frac{d}{dx} - 1)^n x^n,$$

(13)

a class of polynomials that are orthogonal over the interval $[0, \infty)$ with respect to the weight function $g(x) = e^{-x}$. The weight for computing the quadrature is computed as

$$w_i = \frac{x_i}{(n+1)^2[L_{n+1}(x_i)]^2}.$$

(14)

While the computation for $x_i$ and $w_i$ is complicated, in practice we can use a look up table to store corresponding $x_i$ and $w_i$ for a given $n$.

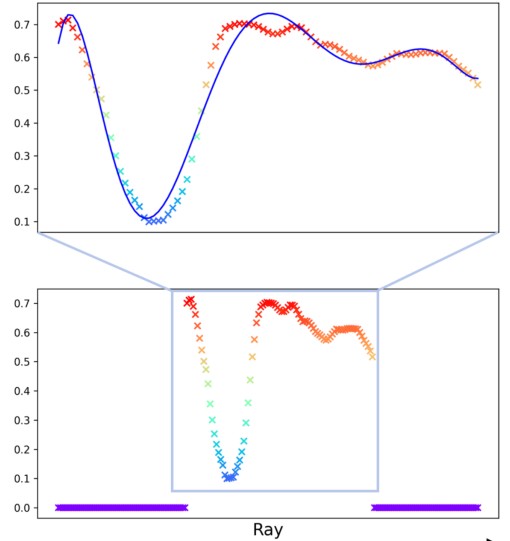

Figure 2: Verification on using the Gauss-Laguerre quadrature for volume rendering. We plot the red channel of the color function w.r.t. the ray it corresponds to. The color function remains zero in most of the interval (bottom). We use a 7th-degree polynomial to approximate the non-zero region (top). As can be seen, the color function itself is similar to a polynomial, validating the use of our approach.

## 4    GL-NeRF

We developed our algorithm based on a simple observation of the integral for volume rendering. Eq. 1 is an approximation to the integral

$$C(\boldsymbol{r}) = \int_{t_n}^{t_f} T(t)\sigma(\boldsymbol{r}(t))\boldsymbol{c}(\boldsymbol{r}(t), d)dt,$$

(15)

where

$$T(t) = exp(-\int_{t_n}^{t} \sigma(\boldsymbol{r}(s))ds).$$

(16)

### 4.1    Volume rendering and Gauss-Laguerre quadrature

Let

$$x(t) = \int_{t_n}^{t} \sigma(\boldsymbol{r}(s))ds, \qquad (17)$$

we have

$$\frac{dx}{dt} = \sigma(\boldsymbol{r}(t)). \qquad (18)$$

Since $\sigma(\boldsymbol{r}(t)) \geq 0$, $x(t)$ is a monotonically non-decreasing function of $t$, therefore, $x$ has a unique correspondence with $t$ on increasing intervals. With this observation, we can do a change of variables for Eq. 15 to get

$$
\begin{aligned}
C(\boldsymbol{r}) &= \int_{t_n}^{t_f} T(t)\sigma(\boldsymbol{r}(t))\boldsymbol{c}(\boldsymbol{r}(t), d)dt \\
&= \int_{t_n}^{t_f} e^{-x}\boldsymbol{c}(\boldsymbol{r}(t), d)\frac{dx}{dt}dt \qquad (19) \\
&= \int_{x(t_n)}^{x(t_f)} e^{-x}\boldsymbol{c}(\boldsymbol{r}(x), d)dx.
\end{aligned}
$$

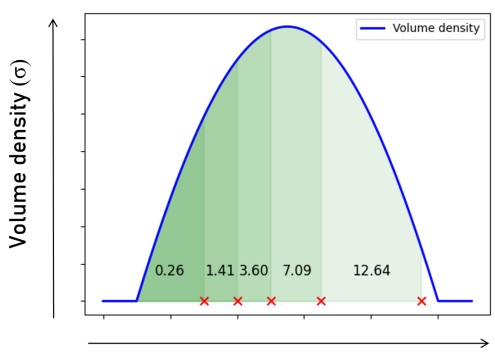

Figure 3: **Point Selection strategy in GL-NeRF.** We choose points along the ray that satisfy the integral from zero to the point of the volume density function should be equal to the roots of Laguerre polynomials. The points selected is then used for querying the color. In the figure above is an example of choosing 5 points using a 5-degree Laguerre polynomial. The number on the plot indicates the value of the integral from zero to the right boundary of the region.

As can be seen from Eq. 19, the integral for volume rendering is a weighted integral of $\boldsymbol{c}(\boldsymbol{r}(x), d)$ with the weight function to be $g(x) = e^{-x}$. We can extend the integral interval from $[x(t_n), x(t_f)]$ to $[0, \infty)$ since the integral between $[0, x(t_n))$ and $(x(t_f), \infty)$ are zero. Thus, we have

$$C(\boldsymbol{r}) = \int_{0}^{\infty} e^{-x}\boldsymbol{c}(\boldsymbol{r}(t(x)), d)dx, \qquad (20)$$

a pure exponentially weighted integral with respect to the color function, which is of the exact same form as required by the Gauss-Laguerre quadrature.

### 4.1.1 Gauss-Laguerre quadrature for volume rendering

As discussed in Sec. 3, the Gauss-Laguerre quadrature guarantees the highest algebraic precision when computing integral over polynomials. To perform the Gauss-Laguerre quadrature for volume rendering integral calculation, a natural question arises: **is the color function a polynomial, or can it be approximated by a polynomial with a satisfactory error rate?**

To answer this question, we first analytically give out a fundamental theorem, and then empirically approximate the color function with polynomials.

**Theorem 4.1** (Stone-Weierstrass theorem). *Suppose $f$ is a continuous real-valued function defined on the real interval $[a, b]$. For every $\epsilon > 0$, there exists a polynomial $p$ such that for all $x$ in $[a, b]$, we have $|f(x) - p(x)| < \epsilon$.*

Since the pixel color is contributed by points that have a density larger than a threshold (*i.e.* regions near the surface), we can overlook the points in the empty space and only analyze the remaining part of the color function. In Fig. 2 we plot a representative of how the color function looks like. As can be seen from the figure, it has a major region with values greater than zero while the others remain zero. When approximating the non-zero region with a 7-th degree polynomial, we have a relative error rate lower than $6.5\%$. While the relative error is not sufficiently low, we argue that we can increase the degree to better approximate it since it's cintinuous by nature. On the other hand, this specific landscape is fluctuated and for most of the cases, the error rate could be smaller than $1\%$. This suggests that Theorem 4.1 holds in our case, thus the Gauss-Laguerre quadrature can be used for computing the volume rendering integral.

### 4.1.2 Point selection in GL-NeRF

Different from NeRF's sampling strategy, the Gauss-Laguerre quadrature enables us to use a deterministic point selection strategy for the color samples. Recall Eq. 17 is the integral variable

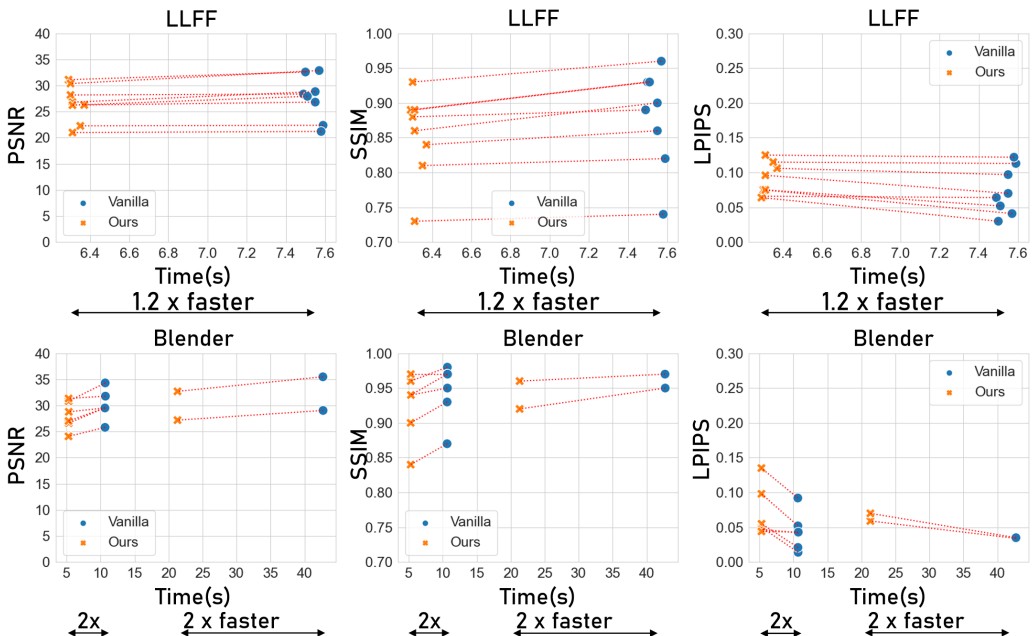

Figure 4: Comparison between GL-NeRF and vanilla NeRF in terms of render time and quantitative metrics. Each point on the figure represents an individual scene. We showcase that with the drop of computational cost GL-NeRF provides, the average time needed for rendering one image is 1.2 to 2 times faster than the vanilla NeRF. In the mean time, the overall performance remains almost the same despite some minor decreases.

for Eq. 20. This means if we want to use the Gauss-Laguerre quadrature to approximate Eq. 20, we have to choose points $x_i$ that are the root of $n$th-degree Laguerre polynomials. Since every $x_i$ has a corresponding $t_i$ following Eq. 17, we can choose $t_i$ based on given value of $x_i$, as depicted in Fig. 3. Specifically, we want the integral Eq. 17 to be equal to the roots of an $n$th-degree Laguerre polynomial. Fig. 3 gives an example of $n = 5$. In the figure, the numbers in the five regions filled with different colors indicate the integral value of the volume density function from zero to the right boundary of the regions. A pseudocode for GL-NeRF rendering is shown in Algo. 1.

### 4.1.3 Intuitive understanding of the points selected using the Gauss-Laguerre quadrature

Since the points near the surface contribute the most to the final color of the pixel as discussed in [20, 29, 31], the optimal point selection strategy should choose points near the surface. The volume density, on the other hand, increases remarkably near the surface and remains close to zero at other areas. Therefore, the integral value of it Eq. 17 should also increases significantly near the surface and remains almost unchanged throughout the rest of the space. Therefore, most of the points chosen using GL-NeRF should lie around the surface of the underlying scene. Consider a case when $n = 8$, we want to choose points $t_i, i = 1, 2, \ldots, 8$ such that $x(t_i)$ in Eq. 17 should be equal to the value $x_i$ given in the look-up table Tab. 1. Notice that the first few value for $x_i$ (say first three) are small so that they could be reached by the integral of volume density near the surface easily. These values have relatively larger weights assigned to them. Evaluating the color of these points using a neural network and summing them up using the weights $w_i$ given in Eq.

| $x_i$ | $w_i$ |
|-------|-------|
| 0.17 | $3.69 \times 10^{-1}$ |
| 0.90 | $4.19 \times 10^{-1}$ |
| 2.25 | $1.76 \times 10^{-1}$ |
| 4.27 | $3.33 \times 10^{-2}$ |
| 7.05 | $2.79 \times 10^{-3}$ |
| 10.76 | $9.08 \times 10^{-5}$ |
| 15.74 | $8.49 \times 10^{-7}$ |
| 22.86 | $1.05 \times 10^{-9}$ |

Table 1: Gauss-Laguerre quadrature look-up table when $n = 8$.

1 following Eq. 12 would contribute mostly to the pixel color. Notice that even though the last few $x_i$ are quite large and may not be reached by Eq. 17 along the ray, their corresponding weights are so small that they almost couldn't affect the final result of the pixel color. Hence, the points selected

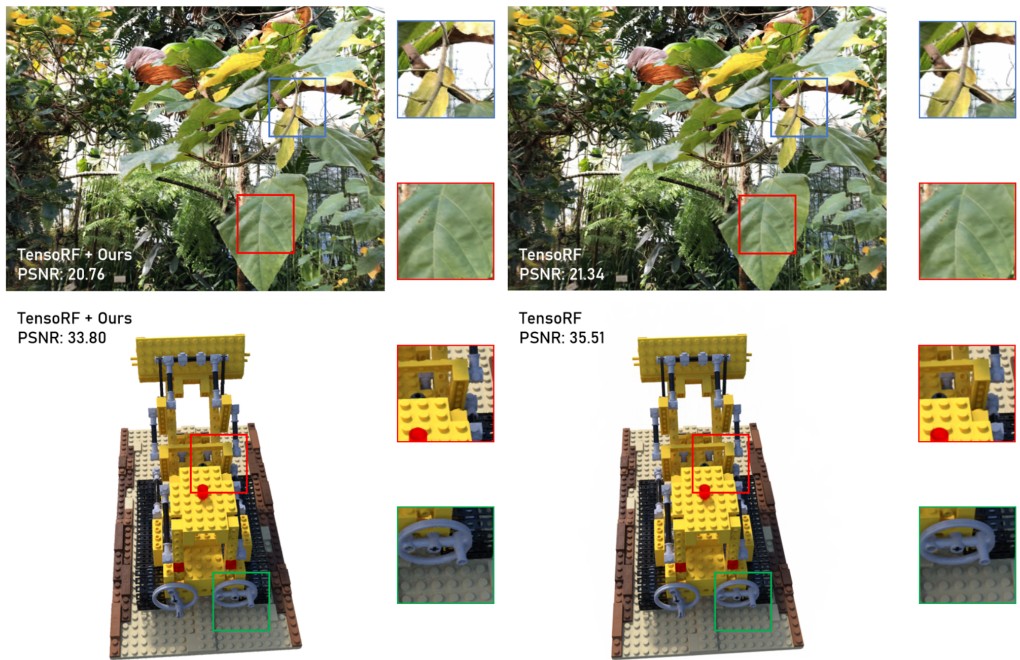

Figure 5: Qualitative results on LLFF (top) and NeRF-Synthetic (bottom) datasets. We could tell from the comparisons that the drop in performances has minimal effect on the visual quality.

| Dataset | Methods | Avg. MLPs↓ | PSNR↑ | SSIM↑ | LPIPS↓ |
|---|---|---|---|---|---|
| LLFF | TensoRF | 118.51 | 26.51 | 0.832 | 0.135 |
| | ours | 4 | 25.63 | 0.797 | 0.146 |
| NeRF-Synthetic | TensoRF | 31.08 | 32.39 | 0.957 | 0.032 |
| | ours | 4 | 30.99 | 0.945 | 0.048 |

Table 2: Quantitative comparison. We demonstrate that our method has a minimal performance drop while significantly reducing the number of color MLP calls.

using GL-NeRF also correspond to the points near the surface, like in previous works [20, 29, 31] that design different neural networks for estimating the surface position, but only without any additional neural networks. Therefore, thanks to the nice property of the Gauss quadrature, ideally we can select the optimal points for computing volume rendering integral if the volume density estimation is oracle.

## 5  Experiments

**Datasets and evaluation metrics.**   We evaluate our method on the standard datasets: NeRF-Synthetic and Real Forward Facing Dataset(LLFF) [26] as in [27] with two different models, *i.e.* Vanilla NeRF [27], TensoRF [6] and InstantNGP [28]. Since our method is training-free, we conduct render-only experiments with the vanilla volume rendering method and our method. We plot the standard render quality evaluation metrics PSNR, SSIM [42] and LPIPS [48] with respect to the average time needed for rendering one image for each scene in Vanilla NeRF. We also report the metrics with averaged color MLP calls for TensoRF and InstantNGP. For Vanilla NeRF, we use 32 points for our method while the network is trained with more than 100 points. For TensoRF and InstantNGP, the results are produced with 4 MLP calls if not otherwise mentioned. More details can be found in Sec. A.1.

### 5.1  Comparison with baselines

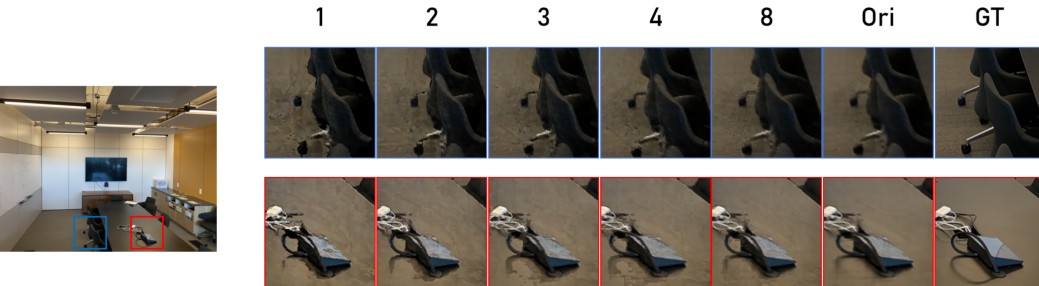

Figure 6: Effect of sample number. The first five columns correspond to the number of sampled points on top. The sixth column shows the result of the original sampling strategy adopted in TensoRF (Ori). The last column is the ground truth visualization of the details in the scene. Our method could achieve comparable results using only 4-8 points while the original strategy requires more than 100 points. The blurriness in the first two columns is inherently the inaccuracy of piece-wise constant density estimation.

| Dataset | Methods | PSNR↑ | SSIM↑ | LPIPS↓ |
|---|---|---|---|---|
| LLFF | Vanilla | 27.62 | 0.88 | 0.073 |
| | ours | 27.21 | 0.87 | 0.087 |
| NeRF-Synthetic | Vanilla | 30.63 | 0.95 | 0.037 |
| | ours | 29.18 | 0.93 | 0.056 |

Table 3: Quantitative comparison when training with GL-NeRF. Vanilla refers to the vanilla NeRF and its sampling strategy while ours refers to replacing the fine sample stage in vanilla NeRF with our sampling strategy, i.e. GL-NeRF. The result for Vanilla NeRF is produced by rendering using more than 100 points while GL-NeRF only uses 32 points.

We showcase that our method can be used for rendering novel views based on pretrained NeRF without further training. We plotted the quantitative metrics of GL-NeRF and original NeRF for an intuitive comparison in Fig. 4. It shows that our method achieves comparable results as the original NeRF while requiring less computation, leading to 1.2 to 2 times faster rendering. We also observed a drop in memory usage due to the fewer MLP calls we have. We further implement our method with TensoRF [6]. As can be seen from Tab. 5, our method significantly reduces the number of MLP calls needed for volume rendering while the rendering quality only drops a little. We observe that the minimal drop in the performance has little effect on the quality of the image. Some qualitative comparisons can be found in Fig. 5. Other than TensoRF, we implement our method on top of InstantNGP [28] to showcase the plug-and-play attribute of our method. Our method performs similarly on Blender dataset to InstantNGP as shown in Tab. 5.

Table 4: Ablation study on the number of points sampled. The more points we have, the better the performance will be. With 8 points, our method is comparable to the original sampling strategy in TensoRF.

| Point number | PSNR↑ | SSIM↑ | LPIPS↓ |
|---|---|---|---|
| 1 | 23.49 | 0.752 | 0.166 |
| 2 | 24.90 | 0.782 | 0.142 |
| 3 | 25.38 | 0.791 | 0.145 |
| 4 | 25.63 | 0.797 | 0.146 |
| 8 | 26.10 | 0.812 | 0.142 |
| Ori | 26.51 | 0.832 | 0.135 |

## 5.2 Discussion on acceleration

The reason why the speed-up in Vanilla NeRF doesn't lead to real-time performance is that it has another heavy neural network for estimating the volume density. While our method needs cheap density estimation, it can be easily achieved by recent efforts in NeRF like factorized tensors [6]. Therefore, reducing the number of color MLP calls needed could lead to real-time performance as shown by previous work [13]. We therefore follow MC-NeRF [13] and develop a real-time renderer based on WebGL

| Blender | | Avg. | Chair | Drums | Ficus | Hotdog | Lego | Mat. | Mic | Ship |
|---|---|---|---|---|---|---|---|---|---|---|
| PSNR↑ | InstantNGP | 32.05 | 34.13 | 25.61 | 31.91 | 36.32 | 34.72 | 29.09 | 34.92 | 29.73 |
| | Ours | 30.35 | 33.08 | 25.07 | 30.13 | 34.78 | 33.05 | 26.54 | 33.02 | 27.15 |

Table 5: Per-scene results on Blender dataset between InstantNGP and ours. We demonstrate that GL-NeRF is able to be plugged into ANY NeRF models.

| Method | PSNR↑ | SSIM↑ | LPIPS↓ | FPS↑ |
|---|---|---|---|---|
| TensoRF | 33.28 | 0.97 | 0.016 | 5.84 |
| ours | 33.09 | 0.97 | 0.016 | 22.34 |

Table 6: Comparison between our method and TensoRF on Lego scene using WebGL-based renderer. The result is collected from an AMD Ryzen 9 5900HS CPU. GL-NeRF is able to provide almost real-time rendering while remaining similar quality as TensoRF.

and train a small variant of TensoRF with $8$ channels for each density component and color component and 32 as hidden size for the color MLP. The result on Lego in the Blender dataset is shown in Tab. 5.4. GL-NeRF is able to provide almost real-time performance in WebGL with similar quality as TensoRF running on an AMD Ryzen 9 5900HS CPU thanks to the reduced number of color MLP calls.

### 5.3 Ablation studies

We further study the effect of sampled points per ray. We conduct experiments using the TensoRF model on the LLFF dataset. We found that 8 points per ray already shows comparable results to the original sampling strategy that uses more than 100 points. Quantitative comparison can be found in Tab. 4. Qualitatively, in Fig. 6 we found that less number of points would lead to blurrier results. Since the points selected using GL-NeRF intuitively correspond to where the surface is, we argue that the blurriness comes from the inherent inaccuracy of piece-wise constant density estimation.

### 5.4 Discussion on GL-NeRF usage for training

While we mainly showcase that GL-NeRF is a general alternative to the sampling strategy for volume rendering at test time, it is also capable of being used for training. We demonstrate this by replacing the fine sample stage in Vanilla NeRF with GL-NeRF and show the result in Tab. 5 and Tab. 9. GL-NeRF is able to produce on-par results with the vanilla sampling strategy but use a much smaller number of points, i.e. 32 for GL-NeRF and more than 100 for vanilla NeRF.

## 6 Conclusion

In this paper, we propose GL-NeRF, a novel approach for calculating the volume rendering integral. We show that with a simple change of variable, the Gauss-Laguerre quadrature can be used for computing the volume rendering integral. Thanks to the highest algebraic precision guaranteed by the Gauss-Laguerre quadrature, GL-NeRF significantly reduces the number of MLP calls needed for the volume rendering integral. We justify the use of the Gauss-Laguerre quadrature theoretically and empirically and showcase the plug-and-play attribute of GL-NeRF in two different NeRF models. Experiments show the potential of GL-NeRF being used for accelerating any existing NeRF model. We also demonstrate that GL-NeRF can be used for training vanilla NeRF, providing a potential new direction for neural rendering research.

**Limitations.** While GL-NeRF shows promising results in reducing the number of MLP calls needed, it still affects the rendering quality despite the theoretical guarantee of the highest precision. How to improve the performance so that it would meet the theoretical results would be interesting.

## Acknowledgement

The authors would like to thank the author of MC-NeRF for the useful tips on developing the WebGL renderer. This work has been funded in part by the Army Research Laboratory (ARL) award W911NF-23-2-0007, DARPA award FA8750-23-2-1015, and ONR award N00014-23-1-2840.

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

# A   Appendix

Here we introduce the implementation details, give a brief introduction of the Gauss-Laguerre quadrature and present our quantitative results on NeRF-Synthetic and LLFF datasets.

## A.1   Implementation details

For Vanilla NeRF, our experiments are conducted based upon NeRF-PyTorch [45], a reproducible PyTorch implementation of the original NeRF [27]. We implement our method by changing the hierarchical sampling strategy into our point selection method using the Gauss-Laguerre quadrature. We follow the standard setting as done in [27] to train a "coarse" and a "fine" network for evaluation. We use a learning rate of $5 \times 10^{-4}$ that exponentially decays to $5 \times 10^{-5}$ over the course of optimization. Each scene is trained for 200k iterations using a single NVIDIA RTX 6000 GPU. We use 128 coarse samples and 32 fine samples to test our method. Since the density estimation for "coarse" and "fine" network are not aligned, we test our method using only the "fine" network by first using "coarse" samples to query it for an estimation of density, then use GL-NeRF to select 32 points for final rendering as discussed in Sec. 4.1.2. LLFF scenes are trained and tested with 64 coarse samples and 64 fine samples for baseline method, while NeRF-Synthetic scenes require 128 coarse samples and 64 fine samples. For TensoRF, we directly use the pretrained checkpoints in the folder VM48 provided by the authors. The qualitative results are produced by 4 neural network calls if not otherwise mentioned. For InstantNGP, we build our code on top of the public PyTorch implementation [12] and train it with the default setting. The final results for GL-NeRF are also produced by 4 neural network calls.

## A.2   Gauss-Laguerre quadrature

The Gauss-Laguerre quadrature is an approximation formula for computing integrals over the semi-infinite interval $[0, +\infty)$ with the weight function $e^{-x}$ and reads as

$$\int_0^{+\infty} e^{-x} f(x) dx \approx \sum_{k=0}^{n} w_k f(x_k). \tag{21}$$

Here $x_0, x_1, \cdots, x_n \in [0, +\infty)$ are the zeros of the Laguerre polynomial $L_{n+1} = L_{n+1}(x)$ of degree $(n+1)$:

$$L_{n+1}(x) = \frac{1}{(n+1)!} e^x \frac{d^{n+1}}{dx^{n+1}} (x^{n+1} e^{-x}),$$

for $n = -1, 0, 1, \cdots$, and the coefficients

$$w_k = \frac{1}{x_k [L'_{n+1}(x_k)]^2}, \quad k = 0, 1, 2, \cdots, n. \tag{22}$$

From the Leibniz formula, it is easy to see that $L_n(x)$ is a polynomial of degree $n$ and the coefficient of $x^n$ is $\frac{(-1)^n}{n!}$. In particular, we have

$$L_0 = 1, \qquad L_1 = 1 - x, \qquad L_2 = \frac{1}{2} x^2 - 2x + 1, \cdots.$$

The fundamental property of the Laguerre polynomials is

**Theorem A.1.** *The Laguerre polynomials $L_n = L_n(x)$ are orthogonal with respect to the weight function $e^{-x}$, that is,*

$$\int_0^{+\infty} e^{-x} L_n(x) L_m(x) dx = \begin{cases} 0, & n \neq m, \\ 1, & n = m. \end{cases}$$

*Proof.* Assume $m \leq n$ and set $g_k(x) = x^k e^{-x}$. From the Leibniz formula it follows that, for $j < k$, $g_k^{(j)}(x)$ is a product of $xe^{-x}$ and a polynomial of degree $(k-1)$ and thereby $g_k^{(j)}(0) = 0 = g_k^{(j)}(+\infty)$

for $j < k$. Thus, we deduce that

$$n!m! \int_0^{+\infty} e^{-x} L_n(x) L_m(x) dx$$

$$= \int_0^{+\infty} e^{-x} e^x g_n^{(n)}(x) e^x g_m^{(m)}(x) dx$$

$$= \int_0^{+\infty} e^x g_m^{(m)}(x) dg_n^{(n-1)}(x)$$

$$= g_n^{(n-1)}(x) [e^x g_m^{(m)}(x)] \big|_0^{+\infty}$$

$$\quad - \int_0^{+\infty} [e^x g_m^{(m)}(x)]' dg_n^{(n-2)}(x)$$

$$= - g_n^{(n-2)}(x) [e^x g_m^{(m)}(x)]' \big|_0^{+\infty}$$

$$\quad + \int_0^{+\infty} [e^x g_m^{(m)}(x)]'' dg_n^{(n-3)}(x)$$

$$= \cdots\cdots$$

$$= (-1)^n \int_0^{+\infty} g_n(x) [e^x g_m^{(m)}(x)]^{(n)}(x) dx \tag{23}$$

By the Leibniz formula, we have

$$\left[ e^x g_m^{(m)}(x) \right]^{(n)} = \sum_{j=0}^n \frac{n!}{(n-j)!j!} (e^x)^{(n-j)} g_m^{(m+j)}(x)$$

$$= e^x \sum_{j=0}^n \frac{n!}{(n-j)!j!} g_m^{(m+j)}(x)$$

and thereby

$$n!m! \int_0^{+\infty} e^{-x} L_n(x) L_m(x) dx$$

$$= (-1)^n \sum_{j=0}^n \frac{n!}{(n-j)!j!} \int_0^{+\infty} x^n g_m^{(m+j)}(x) dx$$

$$= \sum_{j=0}^{n-m} \frac{n!(-1)^{n+m+j}}{(n-j)!j!} \int_0^{+\infty} \frac{n! x^{n-m-j}}{(n-m-j)!} g_m(x) dx$$

$$= \sum_{j=0}^{n-m} \frac{n!(-1)^{n+m+j}}{(n-j)!j!} \int_0^{+\infty} \frac{n!}{(n-m-j)!} x^{n-j} e^{-x} dx$$

$$= (-1)^n \sum_{j=0}^{n-m} \frac{n!}{(n-j)!j!} (-1)^{m+j} \frac{n!(n-j)!}{(n-m-j)!}$$

$$= (-1)^{n+m} \frac{(n!)^2}{(n-m)!} \sum_{j=0}^{n-m} \frac{(n-m)!}{j!(n-m-j)!} (-1)^j$$

$$= (-1)^{n+m} \frac{(n!)^2}{(n-m)!} (1-1)^{n-m}.$$

Here the second equality is similar to that in 23 and the fourth uses

$$\int_0^{+\infty} x^{n-j} e^{-x} dx = (n-j)!.$$

This completes the proof. □

The orthogonality of the Laguerre polynomials ensures that the $L_n(x)$'s are linearly independent and $L_{n+1}(x)$ has $(n+1)$ distinct zeros $x_0, x_1, \cdots, x_n$ in $[0, +\infty)$ [11]. With the zeros, the coefficients $w_k$ are chosen so that the following $(n+1)$ equalities

$$\int_0^{+\infty} e^{-x} x^j dx = \sum_{k=0}^{n} w_k x_k^j, \tag{24}$$

hold for $j = 0, 1, \cdots, n$. This leads to a system of $(n+1)$ linear algebraic equations for the unknowns $w_k$ and the corresponding coefficient matrix is the Vandermonde matrix $[x_k^j]_{(n+1)\times(n+1)}$. The latter is invertible since the zeros are distinct and therefore the $w_k$'s are uniquely determined. The specific expressions of the $w_k$'s are given in 22 [11].

It is remarkable that all the coefficients $w_k$ are non-negative. This important property ensures the stability and convergence of the Gauss-Laguerre quadrature [11]. Moreover, we have

**Theorem A.2.** *The algebraic precision of the Gauss-Laguerre quadrature 21 is $(2n+1)$ exactly. Namely, "$\approx$" in 21 is "$=$" if $f(x)$ is a polynomial of degree $(2n+1)$ and is not "$=$" if $f(x)$ is a polynomial with degree higher than $(2n+1)$.*

*Proof.* Notice that

$$(n+1)! L_{n+1}(x) = (-1)^{n+1} \Pi_{k=0}^{n}(x - x_k)$$

is a polynomial of degree $(n+1)$. Since

$$0 < \int_0^{+\infty} e^{-x} L_{n+1}^2(x) dx \neq 0 = \sum_{k=0}^{n} w_k L_{n+1}^2(x_k),$$

the precision is less than $2(n+1)$. On the other hand, for any polynomial $p = p(x)$ of degree $(2n+1)$ there are two polynomials of degree $n$ such that $p(x) = q(x) L_{n+1}(x) + r(x)$. Notice that $q(x)$ can be written as a linear combination of $L_0(x), L_1(x), \cdots, L_n(x)$. Compute

$$\begin{aligned}
\sum_{k=0}^{n} w_k p(x_k) &= \sum_{k=0}^{n} w_k [q(x_k) L_{n+1}(x_k) + r(x_k)] \\
&= \sum_{k=0}^{n} w_k r(x_k) \\
&= \int_0^{+\infty} e^{-x} r(x) dx \\
&= \int_0^{+\infty} e^{-x} [q(x) L_{n+1}(x) + r(x)] dx \\
&= \int_0^{+\infty} e^{-x} p(x) dx.
\end{aligned}$$

Here the third equality is due to 24 (the choice of $w_k$) and the fourth is due to the orthogonality of $L_{n+1}(x)$ and $q(x)$ with respect to the weight function. Hence the proof is complete. $\square$

For further details on the Gauss-Laguerre quadrature and for other Gauss quadratures, the interested reader is referred to the book [11].

### A.3 Proof for Theorem 4.1

Here we present a proof of the well-known Stone-Weierstrass theorem, basically from [5]. We rephrase the theorem here for readability.

**Theorem A.3.** *Suppose $f = f(x) : [0, 1] \to (-\infty, \infty)$ is continuous. Then for any $\epsilon > 0$ there is a polynomial $p = p(x)$ satisfying*

$$\sup_{x \in [0,1]} |f(x) - p(x)| < \epsilon.$$

*Namely, polynomials are dense in the Banach space $C[0, 1]$.*

| NeRF-Synthetic | | Avg. | Chair | Drums | Ficus | Hotdog | Lego | Mat. | Mic | Ship |
|---|---|---|---|---|---|---|---|---|---|---|
| PSNR↑ | TensoRF | 32.39 | 34.68 | 25.58 | 33.37 | 36.81 | 35.51 | 29.54 | 33.59 | 30.12 |
| | Ours | 30.99 | 33.98 | 25.15 | 30.41 | 35.75 | 33.80 | 27.32 | 32.52 | 30.12 |
| SSIM↑ | TensoRF | 0.96 | 0.98 | 0.93 | 0.98 | 0.98 | 0.98 | 0.94 | 0.98 | 0.88 |
| | Ours | 0.94 | 0.98 | 0.92 | 0.96 | 0.97 | 0.97 | 0.91 | 0.98 | 0.87 |
| LPIPS↓ | TensoRF | 0.032 | 0.014 | 0.059 | 0.015 | 0.017 | 0.009 | 0.036 | 0.012 | 0.098 |
| | Ours | 0.048 | 0.019 | 0.068 | 0.043 | 0.031 | 0.015 | 0.088 | 0.029 | 0.095 |
| **LLFF** | | Avg. | Fern | Flower | Fortress | Horns | Leaves | Orchid | Room | Trex |
| PSNR↑ | TensoRF | 26.51 | 25.31 | 28.22 | 31.14 | 27.64 | 21.34 | 20.02 | 31.80 | 26.61 |
| | Ours | 25.63 | 24.11 | 27.24 | 30.41 | 26.86 | 20.76 | 18.91 | 30.82 | 25.94 |
| SSIM↑ | TensoRF | 0.83 | 0.82 | 0.86 | 0.89 | 0.86 | 0.75 | 0.66 | 0.95 | 0.89 |
| | Ours | 0.80 | 0.76 | 0.82 | 0.87 | 0.83 | 0.72 | 0.59 | 0.92 | 0.87 |
| LPIPS↓ | TensoRF | 0.135 | 0.161 | 0.121 | 0.084 | 0.146 | 0.167 | 0.204 | 0.093 | 0.108 |
| | Ours | 0.146 | 0.181 | 0.115 | 0.089 | 0.146 | 0.146 | 0.255 | 0.122 | 0.118 |

Table 7: Per-scene quantitative comparison between TensoRF and ours.

*Proof.* Fix $f = f(x) \in C[0,1]$ and $\epsilon > 0$. Since $f = f(x)$ is continuous on the bounded closed interval $[0,1]$, it is bounded and uniformly continuous, meaning that there are positive numbers $M > 0$ and $\delta = \delta(\epsilon) > 0$ such that

$$|f(x)| \leq M, \qquad |f(x) - f(y)| < \frac{\epsilon}{2} \tag{25}$$

for any $x, y \in [0,1]$ satisfying $|x - y| < \delta$.

With $\delta$ fixed, let $n \geq \frac{M}{\delta^2 \epsilon}$ be a positive integer. Consider the $n$th-order Bernstein polynomial [5]

$$p(x) = \sum_{k=0}^{n} f(\frac{k}{n}) C_n^k x^k (1-x)^{n-k}$$

with $C_n^k = \frac{n!}{k!(n-k)!}$. Notice that, for $x \in [0,1]$,

$$C_n^k x^k (1-x)^{n-k} \geq 0, \qquad \sum_{k=0}^{n} C_n^k x^k (1-x)^{n-k} = (x + 1 - x)^n = 1, \tag{26}$$

$$
\begin{aligned}
\sum_{k=0}^{n} k C_n^k x^k (1-x)^{n-k} &= \sum_{k=1}^{n} k \frac{n!}{k!(n-k)!} x^k (1-x)^{n-k} \\
&= nx \sum_{k=1}^{n} \frac{(n-1)!}{(k-1)!(n-1-(k-1))!} x^{k-1} (1-x)^{n-1-(k-1)} \\
&= nx(x+1-x)^{n-1} = nx,
\end{aligned} \tag{27}
$$

$$
\begin{aligned}
\sum_{k=0}^{n} k^2 C_n^k x^k (1-x)^{n-k} &= \sum_{k=0}^{n} k C_n^k x^k (1-x)^{n-k} + \sum_{k=0}^{n} k(k-1) C_n^k x^k (1-x)^{n-k} \\
&= nx + \sum_{k=2}^{n} k(k-1) \frac{n!}{k!(n-k)!} x^k (1-x)^{n-k} \\
&= nx + n(n-1)x^2 \sum_{k=2}^{n} \frac{(n-2)!}{(k-2)!(n-k)!} x^{k-2} (1-x)^{n-k} \\
&= nx + n(n-1)x^2.
\end{aligned} \tag{28}
$$

| NeRF-Synthetic | | Avg. | Chair | Drums | Ficus | Hotdog | Lego | Mat. | Mic | Ship |
|---|---|---|---|---|---|---|---|---|---|---|
| PSNR↑ | Vanilla | 30.63 | 34.32 | 25.80 | 29.54 | 35.49 | 29.53 | 29.04 | 31.78 | 29.52 |
| | Ours | 28.56 | 30.82 | 24.08 | 26.62 | 32.70 | 28.78 | 27.19 | 31.34 | 27.03 |
| SSIM↑ | Vanilla | 0.95 | 0.98 | 0.93 | 0.97 | 0.97 | 0.95 | 0.95 | 0.97 | 0.87 |
| | Ours | 0.93 | 0.96 | 0.90 | 0.94 | 0.96 | 0.94 | 0.92 | 0.97 | 0.84 |
| LPIPS↓ | Vanilla | 0.042 | 0.014 | 0.052 | 0.021 | 0.034 | 0.042 | 0.035 | 0.044 | 0.092 |
| | Ours | 0.070 | 0.050 | 0.098 | 0.055 | 0.059 | 0.047 | 0.070 | 0.044 | 0.135 |
| **LLFF** | | Avg. | Fern | Flower | Fortress | Horns | Leaves | Orchid | Room | Trex |
| PSNR↑ | Vanilla | 27.62 | 26.82 | 28.37 | 32.59 | 28.83 | 22.38 | 21.20 | 32.87 | 27.93 |
| | Ours | 26.53 | 26.27 | 28.19 | 31.12 | 26.81 | 22.27 | 20.99 | 30.38 | 26.24 |
| SSIM↑ | Vanilla | 0.88 | 0.86 | 0.89 | 0.93 | 0.90 | 0.82 | 0.74 | 0.96 | 0.92 |
| | Ours | 0.85 | 0.84 | 0.88 | 0.89 | 0.86 | 0.81 | 0.73 | 0.93 | 0.89 |
| LPIPS↓ | Vanilla | 0.074 | 0.097 | 0.064 | 0.030 | 0.070 | 0.113 | 0.122 | 0.041 | 0.052 |
| | Ours | 0.090 | 0.106 | 0.066 | 0.064 | 0.096 | 0.115 | 0.125 | 0.075 | 0.075 |

Table 8: Per-scene quantitative comparison between Vanilla NeRF and ours.

Then, for any $x \in [0,1]$, we deduce from (25)–(28) that

$$|f(x) - p(x)| = \left| \sum_{k=0}^{n} [f(x) - f(\frac{k}{n})] C_n^k x^k (1-x)^{n-k} \right|$$

$$\leq \sum_{k=0}^{n} |f(x) - f(\frac{k}{n})| C_n^k x^k (1-x)^{n-k}$$

$$\leq (\sum_{k:|x-k/n|<\delta} + \sum_{k:|x-k/n|\geq\delta}) |f(x) - f(\frac{k}{n})| C_n^k x^k (1-x)^{n-k}$$

$$< (\frac{\epsilon}{2} \sum_{k:|x-k/n|<\delta} + 2M \sum_{k:|x-k/n|\geq\delta}) C_n^k x^k (1-x)^{n-k}$$

$$\leq \frac{\epsilon}{2} + 2M \sum_{k:|x-k/n|\geq\delta} \frac{(nx-k)^2}{n^2\delta^2} C_n^k x^k (1-x)^{n-k}$$

$$\leq \frac{\epsilon}{2} + \frac{2M}{n^2\delta^2} \sum_{k=0}^{n} (nx-k)^2 C_n^k x^k (1-x)^{n-k}$$

$$\leq \frac{\epsilon}{2} + \frac{2M}{n^2\delta^2} (n^2 x^2 - 2nxnx + nx + n(n-1)x^2) = \frac{\epsilon}{2} + \frac{2M}{n\delta^2} x(1-x)$$

$$\leq \frac{\epsilon}{2} + \frac{\epsilon}{2} = \epsilon.$$

This completes the proof. □

## A.4 Quantitative results on NeRF-Synthetic and LLFF datasets

In this part, we present our per-scene quantitative results on NeRF-Synthetic and LLFF datasets in Tab. 8 for Vanilla NeRF and in Tab. 7 for TensoRF. The full result for training using GL-NeRF is presented in Tab. 9

| **Blender** | | Avg. | Chair | Drums | Ficus | Hotdog | Lego | Mat. | Mic | Ship |
|---|---|---|---|---|---|---|---|---|---|---|
| PSNR↑ | Vanilla | 30.63 | 34.32 | 25.80 | 29.54 | 35.49 | 29.53 | 29.04 | 31.78 | 29.52 |
| | Ours | 29.18 | 32.43 | 24.38 | 26.92 | 33.91 | 29.49 | 27.27 | 31.55 | 27.47 |
| SSIM↑ | Vanilla | 0.95 | 0.98 | 0.93 | 0.97 | 0.97 | 0.95 | 0.95 | 0.97 | 0.87 |
| | Ours | 0.93 | 0.97 | 0.91 | 0.94 | 0.96 | 0.95 | 0.92 | 0.97 | 0.84 |
| LPIPS↓ | Vanilla | 0.037 | 0.014 | 0.052 | 0.021 | 0.034 | 0.042 | 0.035 | 0.044 | 0.092 |
| | Ours | 0.056 | 0.029 | 0.087 | 0.050 | 0.052 | 0.038 | 0.065 | 0.046 | 0.122 |
| **LLFF** | | Avg. | Fern | Flower | Fortress | Horns | Leaves | Orchid | Room | Trex |
| PSNR↑ | Vanilla | 27.62 | 26.82 | 28.37 | 32.59 | 28.83 | 22.38 | 21.20 | 32.87 | 27.93 |
| | Ours | 27.21 | 26.63 | 28.05 | 31.93 | 28.05 | 22.35 | 21.12 | 32.51 | 27.01 |
| SSIM↑ | Vanilla | 0.88 | 0.86 | 0.89 | 0.93 | 0.90 | 0.82 | 0.74 | 0.96 | 0.92 |
| | Ours | 0.87 | 0.85 | 0.88 | 0.91 | 0.88 | 0.81 | 0.74 | 0.95 | 0.90 |
| LPIPS↓ | Vanilla | 0.073 | 0.097 | 0.064 | 0.030 | 0.070 | 0.113 | 0.122 | 0.041 | 0.052 |
| | Ours | 0.087 | 0.121 | 0.075 | 0.043 | 0.089 | 0.117 | 0.131 | 0.053 | 0.069 |

Table 9: Quantitative results when training on Blender and LLFF Datasets.

