# OpenReview forum: "GL-NeRF: Gauss-Laguerre Quadrature Enables Training-Free NeRF Acceleration"
_NeurIPS.cc/2024/Conference — NeurIPS 2024 poster_

### Official Review · Reviewer_PEjS · 2024-06-27

**Soundness:** 3
**Presentation:** 4
**Contribution:** 2
**Rating:** 5
**Confidence:** 4

**Summary:**

Volume rendering requires numerical integration for estimating output colors. This work proposes using the Gauss-Laguerre quadrature to reduce the number of samples and improve integration accuracy. The paper demonstrates that this method can be a plug-and-play module for any NeRF model. Experimental results show that with a limited drop in performance, the GL-NeRF can significantly reduce the number of ray samples and MLP calls.

**Strengths:**

1. The perspective of improving quadrature for NeRF is new and interesting.
2. The formulation of using the Gauss-Laguerre quadrature looks good, and the mathematical formulation appears rigorous.

**Weaknesses:**

1. The results show a performance drop of about ~2 PSNR, while the speed improvement is not significant.
2.  The motivation is not strong. Most state-of-the-art NeRF approaches use shallow MLPs or even no MLPs, making the evaluation less expensive. Reducing ray samples does not seem to address a core issue in radiance field research.
3. How the points are selected is unclear. The method requires approximating polynomial coefficients and resolving the roots for $x$ However, since $x$ is a highly non-linear function of $t$, finding the samples  $t$ unavoidably requires root finding along the ray, which does not seem to actually improve accuracy or efficiency.

**Questions:**

1. Why use a look-up table? Will this lead to worse performance?
2. Why did the performance not match TensoRF? Would increasing the number of point samples help?

**Limitations:**

The paper mentions that it has a theoretical guarantee of the highest precision. However, there is no evidence that the current precision matches previous works.

---

> ### Author Rebuttal · Authors · 2024-08-07
>
> We’re thankful for your time and the valuable insights you’ve shared. Your input has significantly advanced our project. In response to your feedback, we proposed a brand new perspective for volume rendering with a strong math foundation and validated it with experiments. We’ll address your concerns below.
>
>
> > The results show a performance drop of about ~2 PSNR, while the speed improvement is not significant.
> Since vanilla NeRF’s speed is bottlenecked by the coarse network, we showcase the significant improvement in speed using TensoRF in the general response and paste it below. Our method achieves comparable quality as the baseline while running almost real-time on a AMD Ryzen 9 5900HS CPU.
>
> | Method         | PSNR  | SSIM | LPIPS | FPS   |
> | -------------- | ----- | ---- | ----- | ----- |
> | TensoRF        | 33.28 | 0.97 | 0.016 | 5.84  |
> | Ours + TensoRF | 33.09 | 0.97 | 0.016 | 22.34 |
>
> > The motivation is not strong. Most state-of-the-art NeRF approaches use shallow MLPs or even no MLPs, making the evaluation less expensive. Reducing ray samples does not seem to address a core issue in radiance field research.
>
> We’d like to highlight that the position of our paper is a ***brand new perspective for volume rendering***, and its ***applicability towards the general NeRF pipeline*** that relies on volume rendering. Therefore, the experiments validate that it can be combined with any existing NeRF pipeline that relies on volume rendering regardless of the underlying representation. GL-NeRF is orthogonal to existing state-of-the-art work in the sense that all other works introduce additional representation such as neural networks and grids while GL-NeRF focuses only on the volume rendering integral itself. We believe it's a promising exploration of volume rendering and would bring insights into the radiance field research.
>
>
> > How the points are selected is unclear. The method requires approximating polynomial coefficients and resolving the roots for $x$. However, since $x$ is a highly non-linear function of $t$, finding the samples $t$ unavoidably requires root finding along the ray, which does not seem to actually improve accuracy or efficiency.
>
> We’d like to point out that we do not need to compute the polynomial coefficients and resolving the roots. All the roots and coefficients needed are well defined and can be looked up online / in numpy. Please refer to the document for the function numpy.polynomial.laguerre.laggause for detail.
>
> As for finding $t$ on the ray, traditional volume rendering uses a piecewise constant PDF for approximating the volume density, and so do we. Therefore, there is no need to find roots along the ray for a non-linear function, we use the same approximation as done in the hierarchical sampling strategy.
>
> > Why use a look-up table? Will this lead to worse performance?
>
> The Gauss-Laguerre Quadrature is a well-known formula in numerical analysis, therefore there is no need to solve for the roots and coefficients on the run. Since using a look-up table is $O(1)$ in terms of time complexity, therefore it will not lead to worse performance.
>
> > Why did the performance not match TensoRF? Would increasing the number of point samples help?
>
> We’d like to first point out that GL-NeRF uses fewer points than TensoRF (4 v.s. ~32), leading to a lower performance with no visual quality drop (our qualitative results are randomly chosen from the images, not cherry-picked). On the other hand, when increasing the number of points used for GL-NeRF, since the predefined Laguerre weights approach to zero and are smaller than machine precision(for 32-bit float, it’s $1.175494 × 10^{-38}$), the performance reaches a bottleneck. Therefore, we focus on the fact that GL-NeRF, with higher precision, can reduce the sample points by a large margin.

---

> > ### Comment · Reviewer_PEjS · 2024-08-09
> > **Response by Reviewer PEjS**
> >
> > Thank you for your explanation. Most of my initial concerns have been satisfactorily addressed, and I now recognize the novelty in the proposed method. Although the current approach may seem somewhat limited in its impact, I believe it holds potential for broader applications in other areas. Given these considerations, I would like to raise my score.

---

> > > ### Author Response · Authors · 2024-08-09
> > > **Thank You for Your Encouragement and Recognition**
> > >
> > > Your recognition of our novelty is truly encouraging. And thank you again for your insightful feedback. We greatly appreciate your positive recommendation!

---

### Official Review · Reviewer_cdYt · 2024-07-12

**Soundness:** 3
**Presentation:** 3
**Contribution:** 3
**Rating:** 8
**Confidence:** 4

**Summary:**

This paper proposes a computational method for volume rendering using Gauss-Laguerre quadrature. In the context of NeRF, volume rendering is performed by evaluating MLPs (or other data structures) at a sequence of query points on a ray and integrating the weighted results. The proposed method reduces the number of evaluations without much performance degradation by computing the integrations using Gauss-Laguerre quadrature. The proposed method has been embedded in vanilla NeRF and TensoRF for validation, and the reviewer reports reductions in computation time and memory usage. The reviewer acknowledges and appreciates the effectiveness of the proposed method and looks forward to future discussions on the generality of the proposed method (other backbones and learning time applications).

**Strengths:**

- As emphasized in the paper, the approach proposed in this paper is a replacement of integral computations, not a learning of sampling or a change in data structure. Therefore, it is applicable and highly available for various NeRF variants that use volume rendering.
- The position of the proposed method in NeRF research is clearly stated, especially the related work is very clearly described as an introduction to the proposed method.
- In the introduction of the method, the reviewer's first question was l.177, and the paper answers the question. This helps the understanding of the paper and increases the credibility of the proposed method.

**Weaknesses:**

- The proposed method results in dense sampling near the surface, but can it handle translucent objects? Intuitively, there seems to be a correlation between density solidity and rendering quality. It would be desirable to discuss for which scenes the proposed method is effective and for which scenes it is not.
- The effectiveness of the proposed method as an integration method for oracle or trained models is certain. In general, however, volume rendering is used for both training and inference. It would be better to discuss whether the proposed method is specialized for inference or whether it could be used for learning as well.
- We expect the availability of the proposed method to be very broad. If possible, the application of the proposed method to NeRF backbones other than vanilla NeRF and TensoRF could be considered to further demonstrate the generality of the proposed method.
- The correspondence between the plots in Figure 4 is not clear. It would be easier to compare the results if the corresponding scenes of vanilla and the proposed method were connected by a line.
- Small Comment: Related works -> Related work

**Questions:**

- Can the proposed method be used for learning? If so, the proposed method could be a very powerful tool. (identical to the description in the Weaknesses section).
- Why are the results for TensoRF not shown in Figure 4?
- The sample points up to 8 are very small compared to the use of more than 100 sample points in vanilla NeRF. Could the performance of vanilla NeRF be exceeded with more sample points? In other words, vanilla NeRF also approximates integration by discretization, and this sampling density is constant. We believe that a comparison of vanilla NeRF and the proposed method in terms of sample points vs. image quality will more convincingly demonstrate the effectiveness of the proposed method.

**Limitations:**

- I agree with the limitations described in the conclusion. This reviewer believes that the proposed method contributes to real-time rendering, but it better to be validated to make this claim in the future.

---

> ### Author Rebuttal · Authors · 2024-08-07
>
> Thank you for dedicating your time and providing such perceptive feedback. Your recommendations have considerably enhanced our work. Based on your comments, we have provide a brand new general framework for computing volume rendering and validate its accessibility on 2 baselines. We will address your concerns below.
>
> > The proposed method results in dense sampling near the surface, but can it handle translucent objects? Intuitively, there seems to be a correlation between density solidity and rendering quality. It would be desirable to discuss for which scenes the proposed method is effective and for which scenes it is not.
>
> Since volume rendering is originally derived for pure volumetric data, it is inherently not suitable for modeling translucent objects. To verify this argument, we conduct experiments using TensoRF on the DexNeRF dataset[5] and report the result here.
>
> | Method         | PSNR  | SSIM | LPIPS |
> | -------------- | ----- | ---- | ----- |
> | TensoRF        | 24.02 | 0.86 | 0.288 |
> | Ours + TensoRF | 23.99 | 0.86 | 0.298 |
>
> The result shows that our approach performs similarly to traditional volume rendering methods. However, it is ideal that the scenes are pure volumetric since TensoRF itself suffers at modeling translucent objects.
>
> > The effectiveness of the proposed method ... is certain. In general, however, volume rendering is used for both training and inference. It would be better to discuss whether the proposed method ... could be used for learning as well.
>
> Since our focus is the training-free aspect of the proposed method, the results of network training with it are not reported in the paper. However, GL-NeRF can be used for learning as well. We conduct experiments combining GL-NeRF with Vanilla NeRF and show the results here. The training time is also reduced by $1.2\times$ to $2\times$.
>
> ---
> Blender
>
> | Method                          | PSNR  | SSIM | LPIPS |
> | ------------------------------- | ----- | ---- | ----- |
> | Vanilla NeRF                    | 30.63 | 0.95 | 0.037 |
> | Ours + Vanilla NeRF (training)  | 29.18 | 0.93 | 0.056 |
> | Ours + Vanilla NeRF (test-only) | 28.56 | 0.93 | 0.070 |
>
> LLFF
>
> | Method                          | PSNR  | SSIM | LPIPS |
> | ------------------------------- | ----- | ---- | ----- |
> | Vanilla NeRF                    | 27.62 | 0.88 | 0.073 |
> | Ours + Vanilla NeRF (training)  | 27.21 | 0.87 | 0.087 |
> | Ours + Vanilla NeRF (test-only) | 26.53 | 0.85 | 0.090 |
> ---
>
> > We expect the availability of the proposed method to be very broad. If possible, the application of the proposed method to NeRF backbones other than vanilla NeRF and TensoRF could be considered to further demonstrate the generality of the proposed method.
>
> Please refer to the general response for the combination of GL-NeRF with Instant NGP.
>
> > The correspondence between the plots in Figure 4 is not clear. It would be easier to compare the results if the corresponding scenes of vanilla and the proposed method were connected by a line.
>
> > Small Comment: Related works -> Related work
>
> Thank you for the construction comments! We’ll modify the figure and writing as suggested.
>
> > Why are the results for TensoRF not shown in Figure 4?
>
> As mentioned in MCNeRF, NVIDIA GPUs have specialized components to accelerate the neural network inferences, therefore evaluating on these devices with more sample points may not cost that much. Therefore we instead implement a WebGL-based renderer as WebGL is a more accessible platform that is agnostic to the underlying hardware[1, 2, 3, 4]. Please find the result in the General Response section.
>
> > The sample points up to 8 are very small compared to the use of more than 100 sample points in vanilla NeRF. Could the performance of vanilla NeRF be exceeded with more sample points? In other words, vanilla NeRF also approximates integration by discretization, and this sampling density is constant. We believe that a comparison of vanilla NeRF and the proposed method in terms of sample points vs. image quality will more convincingly demonstrate the effectiveness of the proposed method.
>
> Thanks for this very interesting point. When increasing the number of points used for GL-NeRF, since the predefined Laguerre weights approach to zero and are smaller than machine precision, the performance reaches a bottleneck. Therefore, we focus on the fact that GL-NeRF, with higher precision, can reduce the sample points by a large margin. An example of the Gauss Laguerre Quadrature look-up table with n=64 is presented here. Notice that when n gets bigger, the weights get extremely small and would become smaller than machine precision (for 32-bit float, it’s $1.175494 × 10^{-38}$). We do have a comparison in terms of sample points vs. image quality with a smaller number of points shown here.
>
> | n    | weight          | x              |
> | ---- | --------------- | -------------- |
> | 1    | $0.0563$ | $0.0224$ |
> | 2    | $0.119$ | $0.118$ |
> | 3    | $0.157$ | $0.290$ |
> | ...  | ...             | ...            |
> | 62   | $1.592\times 10^{-88}$ | $204.672$ |
> | 63   | $2.989\times 10^{-94}$ | $218.032$ |
> | 64   | $2.089\times 10^{-101}$ | $234.810$ |
>
>
> >[1] Gupta, Kunal, et al. "MCNeRF: Monte Carlo rendering and denoising for real-time NeRFs." SIGGRAPH Asia 2023 Conference Papers. 2023.
>
> >[2] Chen, Zhiqin, et al. "Mobilenerf: Exploiting the polygon rasterization pipeline for efficient neural field rendering on mobile architectures." Proceedings of the IEEE/CVF Conference on Computer Vision and Pattern Recognition. 2023.
>
> >[3] Reiser, Christian, et al. "Merf: Memory-efficient radiance fields for real-time view synthesis in unbounded scenes." ACM Transactions on Graphics (TOG) 42.4 (2023): 1-12.
>
> >[4] Yariv, Lior, et al. "Bakedsdf: Meshing neural sdfs for real-time view synthesis." ACM SIGGRAPH 2023 Conference Proceedings. 2023.
>
> >[5] Ichnowski, Jeffrey, et al. "Dex-NeRF: Using a neural radiance field to grasp transparent objects." arXiv preprint arXiv:2110.14217 (2021).

---

> > ### Comment · Reviewer_cdYt · 2024-08-12
> >
> > I appreciate the detailed point-by-point responses from the authors. Especially, the additional results suggest that the proposed method is useful for learning, which is very important result. Although a more detailed verification of the stability of the learning and the dependence on initial values is needed in my opinion, I think this is beyond the scope of the paper.
> >
> > At this point I have no further questions. Given the content of the rebuttal, my current judgment is to increase the score. I will carefully monitor the progress of other discussions and reevaluate if necessary.

---

> ### Author Response · Authors · 2024-08-12
> **Thank You for Your Recognition and Positive Recommendation**
>
> We truly value your acknowledgment of our work and your recognition of our responses. Your insightful feedback and favorable recommendation are deeply appreciated.

---

### Official Review · Reviewer_a3Mk · 2024-07-19

**Soundness:** 3
**Presentation:** 3
**Contribution:** 3
**Rating:** 7
**Confidence:** 5

**Summary:**

This paper focuses on accelerating novel view synthesis using neural radiance fields (NeRF). Unlike previous works that concentrate on designing lightweight networks, this study is motivated by the specific volume rendering formula, which includes a negative exponential term in the integration function. By employing Gauss-Laguerre quadrature, the authors approximate this complex integral operation, thus improving the rendering speed of existing NeRFs. This approach is validated on two backbones: the original NeRF and TensoRF, demonstrating speed improvements ranging from 1.2X to 2X.

**Strengths:**

1. The idea is intriguing and represents a promising exploration originating from the specific volume rendering formula.
2. This paper is highly theoretical.

**Weaknesses:**

see Questions.

**Questions:**

In the experiment, the authors validated the proposed method on two backbones (the original NeRF and the TensoRF). However, neither of these represents the current fastest method. It is suggested to compare the proposed method with Instant NGP, DVGO, or other faster alternatives. Such comparisons could not only better verify the plug-and-play capability but also significantly enhance the impact of the paper.

**Limitations:**

n/a.

---

> ### Author Rebuttal · Authors · 2024-08-07
>
> We appreciate the time and thoughtful feedback you've given. Your suggestions have greatly improved our work. Considering your insights, we have provided a highly theoretical framework for calculating volume rendering and validated it using 2 baselines. We will address your concerns below.
>
> >In the experiment, the authors validated the proposed method on two backbones (the original NeRF and the TensoRF). However, neither of these represents the current fastest method. It is suggested to compare the proposed method with Instant NGP, DVGO, or other faster alternatives. Such comparisons could not only better verify the plug-and-play capability but also significantly enhance the impact of the paper.
>
>
> Since Instant NGP is a representative of grid-based representation for volume density, we combined GL-NeRF with Instant NGP as suggested and compared between the versions. We believe doing so would provide a more comprehensive perspective for the plug-and-play attribute of GL-NeRF. As can be seen from the table, our method achieves comparable performance with instant NGP using only $\frac{1}{8}$ number of color MLP calls. Since the experiment using TensoRF on WebGL has shown that reducing the color MLP calls can lead to significant speedup, here we simply showcase the color MLP calls needed in the table. The experiment indicates that as long as the underlying radiance fields pipeline relies on volume rendering, GL-NeRF can be an ***off-the-shelf replacement*** for dense sampling / hierarchical sampling, agnostic to the underlying representation.
>
> | Method             | PSNR  | Avg. color MLP calls |
> | ------------------ | ----- | --------------------- |
> | Instant NGP        | 32.05 | 30.90                 |
> | Ours + Instant NGP | 30.35 | 4.00                  |

---

> > ### Comment · Reviewer_a3Mk · 2024-08-12
> >
> > Thank you for your response and additional experiments. All of my concerns have been addressed. It is a good exploration by replacing the classical volume rendering with Gauss-Laguerre quadrature.

---

> ### Author Response · Authors · 2024-08-12
> **Grateful for Your Recognition and Positive Recommendation**
>
> We greatly appreciate your recognition of our work's novelty. We sincerely thank you for your thoughtful feedback and are grateful for your positive recommendation.

---

### Official Review · Reviewer_GERG · 2024-07-19

**Soundness:** 3
**Presentation:** 3
**Contribution:** 2
**Rating:** 6
**Confidence:** 4

**Summary:**

This paper presents a method for reducing the number of color samples needed for volume rendering in Neural Radiance Fields. The method works by applying Gauss-Laguerre quadrature as a replacement for the importance sampling used by some NeRF methods to reduce the number of calls needed for their fine color MLPs. This approach is shown to yield some efficiency improvements on plain NeRF as well as TensoRF models.

**Strengths:**

The mathematical aspects of the paper are detailed and well supported. It is very clear what the method is trying to achieve and how the Gauss-Laguerre formulation is being applied. Generally, the quality of explanation is good and not too hard to follow, even though the math is fairly dense.

Overall, I think this is an original idea with some likely applications in volume rendering.

**Weaknesses:**

The main issue with this paper is that the experiments do little to give an idea of how the proposed method would compare to the current state of the art, which has progressed quite significantly beyond the baselines shown here. While it is quite believable that the GL method outperforms the naive hierarchical sampling of the original NeRF, I have significant doubts about that holding when applied to something more recent like the proposal networks from Mip-NeRF 360 and Zip-NeRF.

Given the weak evaluation, I would lean towards rejecting.

**Questions:**

It is strange that timings are reported for vanilla NeRF but not TensoRF, which would presumably see a larger gain. Is there a reason for this?

I would suggest that the authors try to show quantifiable improvement on a more recent baseline. If there were a substantial run-time speed up for TensoRF, that would be good, but something lime Mip-NeRF 360 or Zip-NeRF would be even better as that would show how GL compares to another method which tries to draw samples near the surface.

**Limitations:**

It is mentioned, but I think the writing could make it a lot more clear that only the color samples are being reduced, as this significantly affects where the method would actually be expected to provide a speedup. As it is, this is quite easy to miss and could lead to misunderstanding if one does not read the method carefully.

I don't think there are any notable concerns with the paper regarding societal impact.

---

> ### Author Rebuttal · Authors · 2024-08-07
>
> We are grateful for your time and insightful comments. Your valuable suggestions have significantly elevated our work. In light of your comments, we have proposed a brand new mathematical perspective for volume rendering and proved its effectiveness on speedup using two baselines. We will address your concerns below.
>
> > The main issue with this paper is that the experiments do little to give an idea of how the proposed method would compare to the current state of the art.
>
> We validated it with 3 baseline approaches (Vanilla NeRF,  TensoRF and InstantNGP in the rebuttal) and show that our method has a plug-and-play attribute. In the experiments, we primarily aim to demonstrate GL-NeRF’s plug-and-play attribute, and can provide comparable performances as well as reducing the sample points, which results in speedup as a free lunch. Moreover, we’d like to point out that our paper’s main contribution is on the mathematical perspective itself.
>
>
> > proposal networks from Mip-NeRF 360 and Zip-NeRF.
>
> While Mip-NeRF 360 focuses on anti-aliasing and Zip-NeRF is its combination with Instant NGP, we instead conduct experiments by combining GL-NeRF with InstantNGP since our focus is on the volume rendering integral itself. The results can be found in the general response and we paste it here for better reference. The result along with the results on Vanilla NeRF and TensoRF indicates that our method can be incorporated into any NeRF pipeline that relies on volume rendering regardless of the underlying representation.
>
> | Method             | PSNR  | Avg. color MLPs calls |
> | ------------------ | ----- | --------------------- |
> | Instant NGP        | 32.05 | 30.90                 |
> | Ours + Instant NGP | 30.35 | 4.00                  |
>
>
> On the other hand, our method needs no training, therefore can be plugged into any existing pipeline, which is the main advantage of our work compared to e.g. the proposal networks from Mip-NeRF 360 and Zip-NeRF.
>
> > It is strange that timings are reported for vanilla NeRF but not TensoRF, which would presumably see a larger gain. Is there a reason for this?
>
> As mentioned in MCNeRF, NVIDIA GPUs have specialized components to accelerate the neural network inferences, therefore evaluating on these devices with more sample points may not cost that much. Therefore we instead implement a WebGL-based renderer as WebGL is a more accessible platform that is agnostic to the underlying hardware[1, 2, 3, 4]. The result is in the General Response section and we paste it here for better reference. As can be seen from the table, our method achieves comparable quality as TensoRF and achieves almost real-time on WebGL running on a AMD Ryzen 9 5900HS CPU ***by only reducing the sampling points***.
>
> | Method         | PSNR  | SSIM | LPIPS | FPS   |
> | -------------- | ----- | ---- | ----- | ----- |
> | TensoRF        | 33.28 | 0.97 | 0.016 | 5.84  |
> | Ours + TensoRF | 33.09 | 0.97 | 0.016 | 22.34 |
>
> >It is mentioned, but I think the writing could make it a lot more clear that only the color samples are being reduced, as this significantly affects where the method would actually be expected to provide a speedup. As it is, this is quite easy to miss and could lead to misunderstanding if one does not read the method carefully.
>
> Thank you for this construction feedback. We'll modify the manuscript to make the point clearer.
>
> >[1] Gupta, Kunal, et al. "MCNeRF: Monte Carlo rendering and denoising for real-time NeRFs." SIGGRAPH Asia 2023 Conference Papers. 2023.
>
> >[2] Chen, Zhiqin, et al. "Mobilenerf: Exploiting the polygon rasterization pipeline for efficient neural field rendering on mobile architectures." Proceedings of the IEEE/CVF Conference on Computer Vision and Pattern Recognition. 2023.
>
> >[3] Reiser, Christian, et al. "Merf: Memory-efficient radiance fields for real-time view synthesis in unbounded scenes." ACM Transactions on Graphics (TOG) 42.4 (2023): 1-12.
>
> >[4] Yariv, Lior, et al. "Bakedsdf: Meshing neural sdfs for real-time view synthesis." ACM SIGGRAPH 2023 Conference Proceedings. 2023.

---

> > ### Author Response · Authors · 2024-08-13
> > **We sincerely appreciate your time and valuable feedback**
> >
> > Dear Reviewer GERG,
> >
> > We sincerely appreciate the time and effort you have dedicated to providing valuable feedback. As the discussion period concludes on Tuesday, August 13, please let us know if there are any remaining questions or if further clarifications are needed. We would be more than happy to provide any additional details.
> >
> > Best Regards,
> >
> > Authors

---

> > > ### Comment · Reviewer_GERG · 2024-08-13
> > >
> > > I don't think any more clarification is required. I remain a bit skeptical about how broadly applicable the approach is given the need for much faster density evaluation than color, but I recognize the theoretical contribution and improved evaluation. As such I will increase my score to weak accept.

---

> > > > ### Author Response · Authors · 2024-08-13
> > > > **Thank You for Your Encouragement and Recognition**
> > > >
> > > > We are truly encouraged by your recognition of our theoretical contribution. Thank you once again for your insightful feedback, we greatly appreciate your positive recommendation!

---

### Author Rebuttal · Authors · 2024-08-07

# General Response
We’d like to thank all the reviewers for their valuable feedback, especially for acknowledging our contributions regarding the theoretical foundation and the effort we made to validate and show the plug-and-play attribute of GL-NeRF. Regarding common concerns among reviewers, we have added some experiments and presented the results to address these concerns.

## TensoRF Speedup Report
As mentioned in MCNeRF[1], NVIDIA GPUs have specialized components to accelerate the neural network inferences. Therefore, on those hardwares evaluating extra MLPs at more sample points may not incur as significant a cost as that on WebGL. Therefore we implemented a WebGL-based renderer and loaded TensoRF into it to test out the speedup GL-NeRF gives us. We follow MCNeRF to train a small TensoRF on LEGO scene in Blender Dataset that could fit in WebGL-based renderer and report its performance here. Our method achieves almost real-time performance in WebGL with similar quality as TensoRF running on a AMD Ryzen 9 5900HS CPU by reducing the sampling points for color MLP.

| Method         | PSNR  | SSIM | LPIPS | FPS   |
| -------------- | ----- | ---- | ----- | ----- |
| TensoRF        | 33.28 | 0.97 | 0.016 | 5.84  |
| Ours + TensoRF | 33.09 | 0.97 | 0.016 | 22.34 |





## Application of GL-NeRF to Other Approaches

To demonstrate the plug-and-play attribute of GL-NeRF to other approaches, we combine GL-NeRF and Instant NGP[2] for evaluation. The results are shown in the table below. ***The reason we choose Instant NGP is that it is another representative of NeRF pipeline that relies on volume rendering and we want to highlight that GL-NeRF can be incorporated into any NeRF pipeline that relies on volume rendering,  agnostic to the underlying representation.*** Since the result on WebGL using TensoRF proved that reducing the number of sample points can lead to significant speedup, here we simply give the MLP calls needed for reference.


| Method             | PSNR  | Avg. color MLPs calls |
| ------------------ | ----- | --------------------- |
| Instant NGP        | 32.05 | 30.90                 |
| Ours + Instant NGP | 30.35 | 4.00                  |


>[1] Gupta, Kunal, et al. "MCNeRF: Monte Carlo rendering and denoising for real-time NeRFs." SIGGRAPH Asia 2023 Conference Papers. 2023.

>[2] Müller, Thomas, et al. "Instant neural graphics primitives with a multiresolution hash encoding." ACM transactions on graphics (TOG) 41.4 (2022): 1-15.

---

### Decision · Program_Chairs · 2024-09-25

**Decision:**

Accept (poster)

**Comment:**

This paper was reviewed by four experts in the field.  Based on the reviewers' feedback, the decision is to recommend the paper for acceptance.  The reviewers did raise some valuable concerns that should be addressed in the final camera-ready version of the paper. The authors are encouraged to make the necessary changes to the best of their ability.    We congratulate the authors on the acceptance of their paper!